# Impact of axial length correction in high intraocular pressure eyes on intraocular lens power calculation

Yuri Kosaka[1], Takashi Kojima [1,2]*, Akeno Tamaoki[1], Yuki Takagi[1], Ayako Sawaki[1], Tsuyoshi Nogami[1,3,4], Tatsushi Kaga[1]

1 Department of Ophthalmology, Japan Community Health Care Organization, Chukyo Hospital, Nagoya, Japan, 2 Nagoya Eye Clinic, Nagoya, Japan, 3 Major of Vision Sciences, Faculty of Health & Medical Sciences, Aichi Shukutoku University, Aichi, Japan, 4 Major of Vision Sciences, Graduate School of Psychology and Medical Sciences, Aichi Shukutoku University, Aichi, Japan

* kojima@sanjogroup.jp

## Abstract

### Purpose

This study aimed to investigate the relationship between high intraocular pressure (IOP) effect on axial length (AL) and postoperative refractive error after cataract surgery.

### Methods

A retrospective study was conducted on cases that underwent cataract surgery for elevated IOP. AL was corrected based on the change in AL per 1 mmHg of IOP, and the predicted refractive error was evaluated at 3 months postoperatively. Preoperative and postoperative optical biometry was performed using the segmented refractive index method. Intraocular lens (IOL) power was calculated using both the SRK/T and Barrett Universal II (BU II) formulas.

### Results

Optical biometry was obtained in 18 cases and 19 eyes before and after cataract surgery. A moderate correlation was observed between the change in IOP and AL (r = 0.687, P = 0.0012), with the change in AL per 1 mmHg of reduction in IOP calculated as 0.005 mm. The mean change in IOP was 38.87 ± 10.80 mmHg (P = 0.001), and the AL \ significantly shortened by 0.18 ± 0.09 mm (P = 0.001). The mean postoperative refractive errors were 0.36 ± 0.59 D with the SRK/T formula, 0.79 ± 0.58 D with the BU II formula, showing significantly more hyperopia (p = 0.001). After AL correction, the predicted refractive error was reduced to −0.08 ± 0.54 D (SRK/T) and 0.27 ± 0.55 D (BU II).

**Data availability statement:** All relevant data are within the paper and its Supporting Information files.

**Funding:** The author(s) received no specific funding for this work.

**Competing interests:** The authors have declared that no competing interests exist.

## Conclusions

In cataract surgery for eyes with elevated IOP, using corrected AL based on the anticipated IOP reduction appears to decrease the predicted refractive error. Additionally, it should be noted that the BU II formula may result in a hyperopic shift in refractive prediction error, even when using the adjusted AL.

## Introduction

High intraocular pressure (IOP) exceeding 40 mmHg can occur in conditions such as acute primary angle-closure glaucoma (PACG), acute primary angle-closure (APAC), a bulging lens, lens dislocation or subluxation, microphthalmia, and uveitis [1]. In cases of PACG and primary angle-closure (PAC), the mainstay treatment involves relieving the pupillary block, often through lens extraction surgery, which is considered the first-line treatment option [1,2].

However, previous studies have shown that simultaneous phacoemulsification and trabeculotomy can lower IOP and shorten axial length (AL), which may influence refractive prediction error [3]. A previous study reported a correlation between IOP and AL change during pharmacological treatment in patients with high IOP, demonstrating that AL decreased by 0.06 mm for every 10 mmHg reduction in IOP [4]. These findings highlight the need for caution when performing cataract surgery in patients with high IOP. Previous studies investigating the impact of pre- and postoperative changes in IOP and AL on refractive prediction error in high IOP cases have utilized composite refractive index-based AL measurement [5].

Optical AL measurements are classified into the "composite refractive indices method" and "segmental refractive indices method." Regarding the composite refractive index method, AL is calculated by dividing the optical path length from the cornea to the retinal pigment epithelium by the average refractive index (composite refractive index), which is derived based on the average composition of each tissue. In contrast, the segmental refractive indices method calculates AL by summing the lengths of individual tissues, obtained using their specific refractive index. The composite refractive indices method has been reported to be prone to showing errors when the lens ratio differs from the expected value. Moreover, the postoperative AL measurements tend to be shorter than preoperative AL measurements, even in normal eyes [6,7].

In this study, we investigated the relationship between preoperative high IOP and postoperative refractive error using segmental AL measurements.

## Materials and methods

This retrospective study included 41 consecutive patients (43 eyes) who underwent cataract surgery for high IOP exceeding 30 mmHg from October 2019 to March 2024 at the Department of Ophthalmology, Japan Community Health Care Organization Chukyo Hospital, Nagoya, Japan. Patients were included if optical biometry had been

performed both preoperatively and postoperatively. The data were accessed for research purposes in March 31, 2024. All data were anonymized before analysis, and the authors did not have access to personally identifiable information at any stage of the study.

The study was approved by the Chukyo Hospital Ethics Committee (Approval No.: 2023063) and conducted retrospectively in accordance with the principles of the Declaration of Helsinki.

First, the IOP changes and the corresponding change in AL per unit change in IOP were calculated based on preoperative and postoperative measurements; the correlation between these two values was then analyzed. Next, the predicted refractive error—defined as the difference between the postoperative subjective spherical equivalent refraction and predicted spherical equivalent refraction—was calculated at 3 months after surgery in all patients who achieved a best-corrected visual acuity (BCVA) of 0.5 or better, measured using decimal visual acuity, and underwent in-the-bag intraocular lens (IOL) fixation. Cases involving predicted refractive error in an eye with keratoconus (1 eye), concomitant vitrectomy (comprising 1 eye with lens dislocation, 1 eye with zonule ligament rupture) were excluded from the study. Additionally, corrected AL values were calculated based on the changes in IOP and AL, and the corresponding predicted refractive error was evaluated.

## Biometrics

Optical biometry was performed using the ARGOS biometer (Alcon Laboratories, Fort Worth, TX) with segmental refractive index method; IOP was measured using iCare TA01i (M.E. Technica, Finland) or TONOREF II (NIDEK, Gamagori, Japan); and corneal shape analysis was conducted using anterior segment optical coherent tomography (AS-OCT) CASIA2 (TOMEY, Nagoya, Japan). The SRK/T and Barrett Universal II (BU II) formulas were used for the IOL power calculation formula. The optimized constants were derived from institutional retrospective outcome data, using postoperative refractive outcomes from eyes implanted with the same IOL models. The parameters used in the SRK/T formula were AL and mean corneal refractive power, while the BU II formula utilized AL, mean corneal refractive power, anterior chamber depth, lens thickness, and corneal diameter.

## Cataract surgery

All patients underwent phacoemulsification through a 3.2-mm temporal corneal incision using an arched knife (Safety handle with arcuate blade), followed by in-the-bag IOL implantation. The implanted IOLs included AN6KA (KOWA) in 8 eyes, CNA0T0 (Alcon) in 3 eyes, SN60WF (Alcon) in 2 eyes, and NX-70S (Santen) in 1 eye.

## Adjustment of axial length

Based on the mean postoperative IOP of $13.15 \pm 2.54$ mmHg, an assumed postoperative IOP of 13 mmHg was used uniformly to estimate the IOP reduction in all cases. The estimated IOP reduction was then multiplied by the change in AL per 1 mmHg decrease IOP to calculate the AL correction value. Finally, the corrected AL was obtained by subtracting the AL correction value from the preoperative AL. In this study, to provide a single correction factor that can be easily applied in clinical practice, the AL correction value was calculated based on the mean postoperative IOP rather than the individual IOP changes for each eye.

## Statistical analyses

The data were analyzed using Instat software for Windows (version 5.0, GraphPad Prism Software, Inc.). The Shapiro–Wilk test was used to assess the data normality, the Spearman rank correlation test was employed to analyze the correlation between IOP and AL, and the Wilcoxon signed-rank test was used to analyze the arithmetic mean and absolute value of the predicted refractive error. A p-value less than 0.05 was considered statistically significant.

## Results

Optical biometry was performed before and after cataract surgery in 19 eyes from 18 patients, comprising 6 male individuals (6 eyes) and 12 female participants (13 eyes), with a mean age of $72.6 \pm 7.7$ years. The study included 14 eyes with acute angle-closure glaucoma, 2 eyes with PACG, and 3 eyes with secondary glaucoma (comprising 1 eye with pseudoexfoliation syndrome, 1 eye with lens dislocation, and 1 eye with uveitis). Combined surgical procedures with phacoemulsification included goniosynechialysis in 16 eyes, trabeculotomy in 1 eye, and vitrectomy in 2 eyes. Additionally, postoperative best-corrected visual acuity of 0.5 or better was achieved in 14 eyes from 13 patients.

### Pre- and post-operative biometric changes

The postoperative AL was significantly reduced, with an average of $0.18 \pm 0.09$ mm (P=0.001) (Table 1). Although the central corneal thickness was significantly greater preoperatively by $33.86 \pm 27.22$ μm (P=0.004), there was no significant difference in mean corneal refractive power before and after surgery (P=0.53). Table 2 presents the keratometric and posterior corneal power measured using AS-OCT. The mean change in corneal refractive power was $-0.10 \pm 0.30$ D for keratometric power (P=0.19) and $0.05 \pm 0.12$ D for posterior power (P=0.07).

### Relationship between intraocular pressure and axial length

The mean reduction in IOP following cataract surgery was $38.87 \pm 10.80$ mmHg (P=0.001), and a moderate correlation was observed between the change in IOP and the change in AL (r=0.687, P=0.0012) (Fig 1). The change in AL per 1 mmHg reduction in IOP was 0.005 mm.

**Table 1. Biometric measurements before and after cataract refractive surgery.**

| Parameters | Pre surgery | Post surgery | Pre-post Difference | *P* value* |
|---|---|---|---|---|
| IOP (mmHg) | $52.02 \pm 11.02$ | $13.15 \pm 2.54$ | $38.87 \pm 10.80$ | <0.001 |
| AL (mm) | $23.02 \pm 1.10$ | $22.83 \pm 1.10$ | $0.18 \pm 0.09$ | <0.001 |
| Average Corneal Refractive Power (D) | $44.81 \pm 1.55$ | $44.77 \pm 1.57$ | $0.03 \pm 0.33$ | 0.53 |
| ACD (mm) | $2.21 \pm 0.35$ | $4.25 \pm 0.23$ | $-2.04 \pm 0.43$ | <0.001 |
| CCT (μm) | $570.43 \pm 30.41$ | $538.54 \pm 30.99$ | $33.86 \pm 27.22$ | <0.004 |
| LT (mm) | $5.19 \pm 0.29$ | $0.80 \pm 0.15$ | $4.39 \pm 0.27$ | <0.001 |
| CD (mm) | $11.57 \pm 0.49$ | $11.66 \pm 0.45$ | $-0.09 \pm 0.54$ | 0.396 |

IOP = Intraocular Pressure; AL = Axial Length; ACD = Anterior Chamber Depth; CCT = Central Corneal Thickness; LT = Lens Thickness; CD = Corneal Diameter.

Data are presented as mean ± standard deviation.

* Wilcoxon signed-rank test.

**Table 2. Comparison of keratometric and posterior corneal power between before and after cataract surgery.**

| Corneal power (D) | Pre surgery | Post surgery | Pre-post Difference (max, min) | *P* value* |
|---|---|---|---|---|
| Keratometric | $44.6 \pm 11.54$ | $44.70 \pm 1.54$ | $-0.10 \pm 0.30$ (0.4, −0.7) | 0.196 |
| Posterior | $-6.31 \pm 0.23$ | $-6.35 \pm 0.25$ | $0.05 \pm 0.12$ (0.20, −0.30) | 0.074 |

Data are presented as mean ± standard deviation.

* Wilcoxon signed-rank test

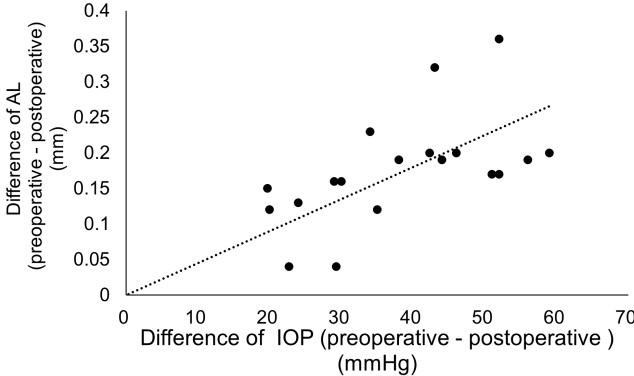

**Fig 1. Correlation between the difference of AL (preoperative – postoperative) (mm) and the difference of IOP (preoperative – postoperative) (mmHg). y = 0.005x (r = 0.687, P = 0.0012).** AL = axial length; IOP = intraocular pressure.

## Predicted refractive errors between the SRK/T and BU II formulas before and after axial length adjustment

As shown in Fig 2, the mean predicted refractive error (MPE) before AL adjustment was 0.36 ± 0.59 D with the SRK/T formula and 0.79 ± 0.58 D with the BU II formula. The BU II formula exhibited a significantly greater hyperopic shift compared to the SRK/T formula (P = 0.001). The median absolute predicted refractive error (MedAPE) was 0.25 D for the SRK/T formula and 0.65 D for the BU II formula, with the BUII formula demonstrating a significantly larger hyperopic error (P = 0.002).

As shown in Fig 3, the MPE after AL adjustment was −0.08 ± 0.54 D for the SRK/T formula and 0.27 ± 0.55 D for the BU II formula, with the BU II formula exhibiting a significantly greater hyperopic shift (P = 0.001). There was no significant difference in the MedAPE between the two formulas (P = 0.63); however, the maximum error was 1.46 D for the BU II formula.

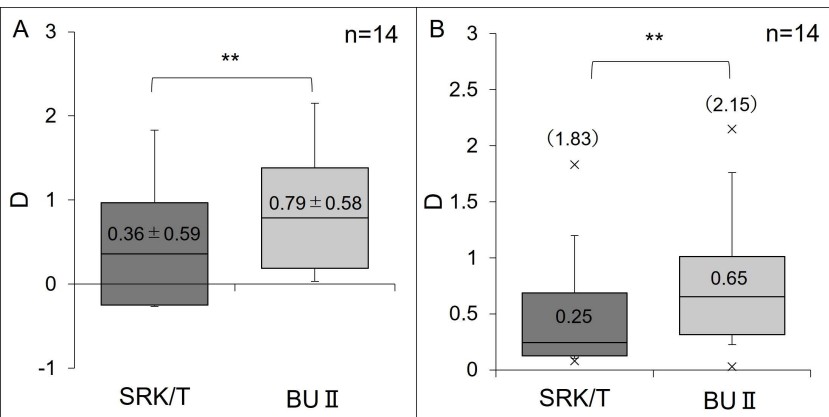

**Fig 2. Predictive refractive error of cataract surgery. (A)** Arithmetic Mean (Mean±SD). **(B)** Absolute Value (Median (Max)). Both the arithmetic means and absolute value showed that BU II formula resulted in significantly more hyperopic compared with the SRK/T formula (P = 0.001). A box-and-whisker plot shows the distribution of data for each group. The line in the center of each box represents the arithmetic mean (A) and the median **(B)**. The bottom and top edges of the box indicate the first and third quartiles, respectively. The whiskers extend to the minimum and maximum values. BU II = Barrett Universal II; SD = standard deviation.

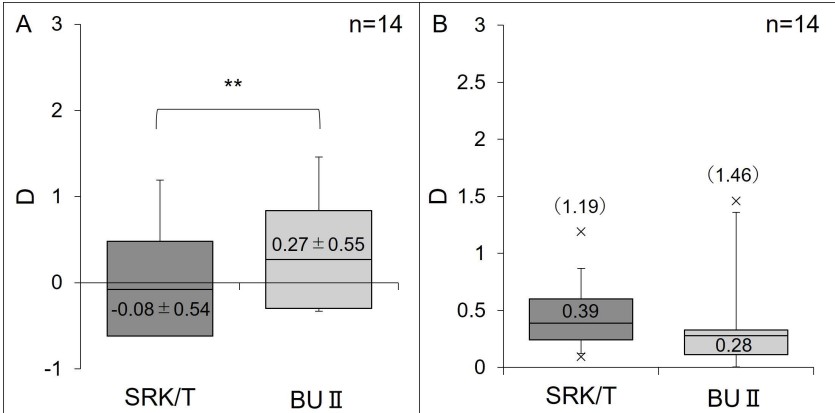

**Fig 3. Predictive refractive error of Predicted Refractive Error After Axial Length Correction. (A)** Arithmetic Mean (Mean±SD). **(B)** Absolute Value (Median (Max)). The BUII formula resulted in significantly more hyperopic outcomes than the SRK/T formula based on the arithmetic mean (P = 0.001). A box-and-whisker plot shows the distribution of data for each group. The line in the center of each box represents the arithmetic mean (A) and the median **(B)**. The bottom and top edges of the box indicate the first and third quartiles, respectively. The whiskers extend to the minimum and maximum values. BU II = Barrett Universal II; SD = standard deviation.

## Discussion

An increase in AL has been reported in eyes with ocular hypertension compared with after IOP reduction, raising concerns about its influence on predicted refractive error in patients undergoing cataract surgery. This study investigated the impact of pre- and post-operative changes in IOP and AL on the predicted refractive error in patients who underwent cataract surgery for preoperative ocular hypertension.

Relative pupillary block and lens bulging are known to occur in PACG and PAC disease [8–11]. Additionally, eyes with elevated IOP due to secondary angle-closure glaucoma may exhibit various anterior segment anatomical characteristics, such as lens vaulting, lens subluxation, and microphthalmia. Numerous studies have reported an association between elevated IOP and AL elongation [3,4,12–17]. A previous study demonstrated that IOP reduction following trabeculectomy resulted in AL shortening, with a positive correlation between the two [17]. Another study reported a strong correlation between IOP reduction and AL shortening, noting that AL decreased by 0.06 mm for every 10 mmHg reduction in IOP [3]. In the present study, a similar correlation was found between changes in IOP and AL; the change in AL per 1 mmHg change in IOP was 0.005 mm, consistent with previous findings.

In normal eyes, AL calculated using the composite refractive index method has been reported to shorten postoperatively. However, AL calculated using the segmented refractive index method shows less change before and after cataract surgery and does not show a significant difference [6,7]. In this study, we utilized an optical biometer capable of measuring AL based on the segmented refractive index method to evaluate eyes with ocular hypertension. We investigated the predicted refractive error in cases wherein AL was measured before and after IOP reduction following cataract surgery.

The effect of elevated IOP on predictive refractive error is possibly attributed to a change in corneal refractive power due to corneal edema. In this study, the central corneal thickness showed an average decrease of 33.86 μm. However, in this study, no significant differences were observed in corneal refractive power before and after the surgery, both in keratometric values and the posterior corneal power. Similarly, a previous study has reported no significant change in corneal radius of curvature before and after cataract surgery in eyes with acute angle-closure glaucoma attacks [18]. These findings suggest that changes in central corneal thickness due to elevated IOP do not affect corneal refractive power.

In the current study, with adjusted AL, the predicted refractive error was reduced from an average of 0.36 to −0.08 D with the SRK/T formula and from an average of 0.79 to 0.27 D with the BU II formula. Furthermore, the BU II formula showed a more hyperopic predicted refractive error compared with the SRK/T formula. In our study cases, the high IOP group had a significantly shallower anterior chamber depth and thicker crystalline lens compared to the cataract group with normal IOP, matched for age and AL (S1 Table). Since the BU II formula has been reported to result in a hyperopic shift with a thicker crystalline lens [19], it was considered that the predicted refractive error was more hyperopically shifted compared to that of the SRK/T formula.

On the other hand, the SRK/T formula tends to produce a myopic shift in eyes with a shallow anterior chamber and short AL, whereas a thicker crystalline lens leads to a hyperopic shift [20]. Therefore, these opposing effects may have counterbalanced each other, resulting in a smaller predicted refractive error after AL adjustment. Although the SRK/T formula utilizes only AL and corneal radius of curvature as input parameters, the BU II formula also incorporates anterior chamber depth and crystalline lens thickness. This suggests that in eyes with ocular hypertension, where the anterior segment anatomy deviates from normal, the predictive accuracy of the BU II formula might be reduced.

This study has some limitations. First, multiple types of tonometers and IOLs were used, owing to the retrospective observational nature of the study. Second, due to the small sample size, there was a mix of cases of different underlying causes of IOP, which might have influenced the results due to differences in the mechanisms of IOP elevation. Further large-scale studies are needed in the future to classify cases based on the mechanisms of IOP elevation and conduct more detailed investigations. Third, multiple types of surgeries were performed simultaneously with cataract surgery in this study. These heterogeneous surgical backgrounds complicate the interpretation of the study results and particularly limit the generalizability of the AL correction coefficient. Additionally, due to the small sample size, more detailed investigations regarding the effects of different underlying pathologies and surgical procedures on intraocular pressure reduction are inherently necessary. Furthermore, previous reports have shown that cases of iris adhesions had a higher possibility of experiencing myopic refractive error after cataract surgery compared to those without iris adhesions [21]. In our study, 84.2% of the cases of iris adhesions underwent goniosynechiolysis, while 5.3% and 10.5% underwent trabeculotomy and vitrectomy, respectively. Increasing the sample size and conducting further investigations focusing solely on cases undergoing combined cataract surgery and goniosynechiolysis is necessary. Fourth, the small sample size is a limitation, and the AL correction formula was not verified in an independent cohort. Future studies are needed to verify the AL correction formula in a separate group of cases of different characteristics.

In conclusion, for cataract surgery in eyes with high IOP, adjusting AL based on the anticipated reduction in IOP effectively reduced the predicted refractive error. Additionally, it should be noted that the BU II formula might still result in a hyperopic shift in refractive prediction error, even when using the adjusted AL.

## Supporting information

**S1 Table. Comparison between normal IOP and high IOP groups with age and axial length matching in self-examination cases.** We compared ACD and LT between the high IOP group in the current study and 57 eyes from 57 patients with normal IOP who underwent cataract surgery alone, matching for age, axial length (AL), and the model of intraocular lens (AN6KA, KOWA). IOP = Intraocular Pressure; ACD = Anterior Chamber Depth; LT = Lens Thickness. * Wilcoxon signed-rank test.
(DOCX)

**S1 Data. Raw data.** This file contains the raw data used in this study, including biometric parameters, corneal power measurements, and data for the normal IOP group.
(XLSX)

## Author contributions

**Conceptualization:** Yuri Kosaka, Takashi Kojima, Akeno Tamaoki, Ayako Sawaki, Tatsushi Kaga.

**Data curation:** Yuri Kosaka, Yuki Takagi, Tsuyoshi Nogami.

**Formal analysis:** Yuki Takagi.

**Funding acquisition:** Tatsushi Kaga.

**Investigation:** Yuri Kosaka, Ayako Sawaki, Tsuyoshi Nogami.

**Methodology:** Takashi Kojima, Akeno Tamaoki.

**Project administration:** Akeno Tamaoki, Yuki Takagi, Tsuyoshi Nogami.

**Resources:** Tatsushi Kaga.

**Supervision:** Takashi Kojima, Akeno Tamaoki.

**Validation:** Takashi Kojima, Akeno Tamaoki, Yuki Takagi, Ayako Sawaki, Tatsushi Kaga.

**Visualization:** Yuri Kosaka, Tsuyoshi Nogami.

**Writing – original draft:** Yuri Kosaka.

**Writing – review & editing:** Takashi Kojima, Akeno Tamaoki, Yuki Takagi, Ayako Sawaki, Tsuyoshi Nogami, Tatsushi Kaga.

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
