## [Decision Letter · Decision Letter 0]

3 Feb 2026

PONE-D-25-53959Impact of axial length correction in high intraocular pressure eyes on intraocular lens power calculationPLOS One

Dear Dr. Kojima,

Thank you for submitting your manuscript to PLOS ONE. After careful consideration, we feel that it has merit but does not fully meet PLOS ONE’s publication criteria as it currently stands. Therefore, we invite you to submit a revised version of the manuscript that addresses the points raised during the review process.

We look forward to receiving your revised manuscript.

Kind regards,

Nader Hussien Lotfy Bayoumi, M.D., FRCS (Glasgow)

Academic Editor

PLOS One

Journal Requirements:

2. We note that there is identifying data in the Supporting Information file “Supporting information table 1.xlsx”. Due to the inclusion of these potentially identifying data, we have removed this file from your file inventory. Prior to sharing human research participant data, authors should consult with an ethics committee to ensure data are shared in accordance with participant consent and all applicable local laws.

-Location data

3.If the reviewer comments include a recommendation to cite specific previously published works, please review and evaluate these publications to determine whether they are relevant and should be cited. There is no requirement to cite these works unless the editor has indicated otherwise.

Reviewer's Responses to Questions

Comments to the Author

1. Is the manuscript technically sound, and do the data support the conclusions?

Reviewer #1: Partly

Reviewer #2: Yes

2. Has the statistical analysis been performed appropriately and rigorously? 

Reviewer #1: Yes

Reviewer #2: Yes

3. Have the authors made all data underlying the findings in their manuscript fully available?

Reviewer #1: Yes

Reviewer #2: Yes

4. Is the manuscript presented in an intelligible fashion and written in standard English?

Reviewer #1: Yes

Reviewer #2: Yes

5. Review Comments to the Author

Reviewer #1: This study investigates the impact of axial length (AL) correction based on anticipated intraocular pressure (IOP) reduction on refractive prediction error after cataract surgery in eyes with high IOP. The topic is clinically relevant, and the use of segmental AL measurement is a strength. However, several methodological issues significantly limit the strength of the conclusions and require clarification and careful discussion.

Key Points for Revision

1.Major Confounding Factor: The primary intervention (cataract surgery/lens extraction) itself is a known cause of AL shortening. The reported AL change (0.005 mm per 1 mmHg IOP drop) is therefore a composite of both the surgical effect and the IOP-lowering effect. The analysis does not adequately control for or disentangle this fundamental confounding. The authors must substantially expand the discussion to explicitly acknowledge this limitation and compare their findings to studies on non-surgical IOP normalization (e.g., Kim et al., 2016).

2.Lack of Methodological Detail: The manuscript states that "optimized IOL constants for each IOL type were applied" but provides no information on how these constants were optimized (e.g., institutional retrospective data, manufacturer's recommendation). This is a critical omission for reproducibility and validation. Please specify the source or method of optimization for the constants used.

3.Small and Heterogeneous Cohort: The study cohort (19 eyes) is small and includes varied concomitant surgical procedures (goniosynechialysis, trabeculotomy, vitrectomy). This heterogeneity may influence the results. The limitations section should more forcefully address how this affects the generalizability of the findings and the proposed AL correction factor.

Reviewer #2: 1.L83: The system used for visual acuity notation needs to be mentioned.

2.It is not clear why the AL correction value was calculated based on the mean postoperative IOP and not based on the change in IOP for each eye.

3.It is confusing to exclude cases of concomitant vitrectomy and zonule ligament rupture (L85) from the study while including an eye with lens dislocation (L120) and two eyes with concomitant vitrectomy (L122).

6. PLOS authors have the option to publish the peer review history of their article (what does this mean?). If published, this will include your full peer review and any attached files.

Do you want your identity to be public for this peer review? For information about this choice, including consent withdrawal, please see our Privacy Policy.

Reviewer #1: No

Reviewer #2: No

---

## [Author Response · Author response to Decision Letter 1]

10 Mar 2026

March 10, 2026

Nader Hussien Lotfy Bayoumi, M.D., FRCS (Glasgow)

Academic Editor

PLOS ONE

Dear Academic Editor:

Re: Revised manuscript, PONE-D-25-53959 Title: Impact of axial length correction in high intraocular pressure eyes on intraocular lens power calculation

The manuscript has been carefully rechecked, and appropriate changes have been made in accordance with the reviewers’ suggestions. We provide below a point-by point response to your and the reviewers’ comments. All changes to our manuscript are indicated in red text.

We thank you and the reviewers for your thoughtful suggestions and insights, from which our manuscript has greatly benefited. We look forward to working with you and the reviewers to move this manuscript closer to publication in PLOS ONE.

Thank you for your consideration. I look forward to hearing from you.

Sincerely,

Takashi Kojima

Department of Ophthalmology, Japan Community Healthcare Organization Chukyo Hospital

1-1-10 Sanjo Minami-ku Nagoya-city, Aichi prefecture, Japan

E-mail: kojima@sanjogroup.jp

We would like to thank the editor and the reviewers for their constructive and insightful comments. We have carefully revised the manuscript to address all concerns. Our point-by-point responses are provided below.

Reviewer 1

We thank the reviewer for the thoughtful comments, which have increased the scientific value of our paper.

Comments to the Author

Comment 1:

Major Confounding Factor: The primary intervention (cataract surgery/lens extraction) itself is a known cause of AL shortening. The reported AL change (0.005 mm per 1 mmHg IOP drop) is therefore a composite of both the surgical effect and the IOP-lowering effect. The analysis does not adequately control for or disentangle this fundamental confounding. The authors must substantially expand the discussion to explicitly acknowledge this limitation and compare their findings to studies on non-surgical IOP normalization (e.g., Kim et al., 2016).

Response1:

We greatly appreciate the reviewer’s insightful comment regarding the potential confounding effect of cataract surgery itself on AL shortening.

To address this concern, we referred to the study by Goto et al. (J Cataract Refract Surg. 2020;46:710–715), which compared the segmental refractive index method and the composite refractive index method for measuring optical AL before and after cataract surgery. Their results revealed that when using the segmental refractive index method, no significant change in AL was observed before and after cataract surgery, suggesting that this measurement method can minimize the impact of AL changes associated with cataract surgery.

On the other hand, a previous study by Kim et al. (2016), which examined changes in the AL associated with nonsurgical IOP lowering intervention, used optical AL measurements based on the composite refractive index method. As the reviewer has accurately pointed out, this method may be more susceptible to confounding effects related to changes in the AL.

Based on these findings, we adopted the segmental refractive index method for optical AL measurement in this study. The rationale for using the segmental refractive index method is described in the Discussion section as follows. “In normal eyes, AL calculated using the composite refractive index method has been reported to shorten postoperatively. However, AL calculated using the segmented refractive index method shows less change before and after cataract surgery and does not show a significant difference [6,7]. In this study, we utilized an optical biometer capable of measuring AL based on the segmented refractive index method to evaluate eyes with ocular hypertension. We investigated the predicted refractive error in cases wherein AL was measured before and after IOP reduction following cataract surgery.” (Discussion, Page 19, Lines 203–209).

Comment 2:

Lack of Methodological Detail: The manuscript states that "optimized IOL constants for each IOL type were applied" but provides no information on how these constants were optimized (e.g., institutional retrospective data, manufacturer's recommendation). This is a critical omission for reproducibility and validation. Please specify the source or method of optimization for the constants used.

Response2:

We greatly appreciate the reviewer for pointing out this important omission. We agree that clarification of the IOL constant optimization process is essential for reproducibility.

We have now described the source and method used for optimizing the IOL constants in the Methods section as follows. “The optimized constants were derived from institutional retrospective outcome data, using postoperative refractive outcomes from eyes implanted with the same IOL models.” (Methods, Page 9, Lines 95–96).

Comment 3:

Small and Heterogeneous Cohort: The study cohort (19 eyes) is small and includes varied concomitant surgical procedures (goniosynechialysis, trabeculotomy, vitrectomy). This heterogeneity may influence the results. The limitations section should more forcefully address how this affects the generalizability of the findings and the proposed AL correction factor.

Response3:

Thank you for your valuable comment. We recognize that the small sample size of 19 eyes and the inclusion of concomitant surgeries such as goniosynechialysis, trabeculotomy, and vitrectomy represent significant limitations of this study.

We have strengthened the Limitations section to clearly state that these additional surgeries may affect postoperative AL and refractive outcomes, and that the generalizability of the proposed AL correction coefficient may therefore be limited. We have added the following sentence to the Discussion section. “Third, multiple types of surgeries were performed simultaneously with cataract surgery in this study. These heterogeneous surgical backgrounds complicate the interpretation of the study results and particularly limit the generalizability of the AL correction coefficient. Additionally, due to the small sample size, more detailed investigations regarding the effects of different underlying pathologies and surgical procedures on intraocular pressure reduction are inherently necessary.” (Discussion, pages 22, lines 239–244).

Reviewer 2

We thank the reviewer for the thoughtful comments that increased the scientific value of our paper.

Comment 1:

L83: The system used for visual acuity notation needs to be mentioned.

Response 1:

Thank you for your valuable comment. We have now clearly specified the visual acuity notation system used in the study in the Methods section as follows. “Next, the predicted refractive error—defined as the difference between the postoperative subjective spherical equivalent refraction and predicted spherical equivalent refraction—was calculated at 3 months after surgery in all patients who achieved a best-corrected visual acuity (BCVA) of 0.5 or better, measured using decimal visual acuity, and underwent in-the-bag intraocular lens (IOL) fixation.” (Page 8, Line 83).

Comment 2:

It is not clear why the AL correction value was calculated based on the mean postoperative IOP and not based on the change in IOP for each eye.

Response 2:

We greatly appreciate your pertinent methodological question. The AL correction value was calculated based on the mean postoperative IOP to derive a single, clinically applicable correction factor that could be easily implemented preoperatively. We have added an explanation regarding this point to the Methods section as follows. “In this study, to provide a single correction factor that can be easily applied in clinical practice, the AL correction value was calculated based on the mean postoperative IOP rather than the individual IOP changes for each eye.” (Pages 10–11, Lines 110–112).

As you rightly pointed out, a correction based on the IOP change for each individual eye may allow for a more personalized estimation. Therefore, we additionally examined this approach and confirmed that the correction values calculated based on individual IOP changes did not materially differ from those obtained using the single correction factor derived from the mean postoperative IOP (13 mmHg).

Comment 3:

It is confusing to exclude cases of concomitant vitrectomy and zonule ligament rupture (L85) from the study while including an eye with lens dislocation (L120) and two eyes with concomitant vitrectomy (L122).

Response 3:

Thank you for your valuable comment. We acknowledge that the description of the inclusion and exclusion criteria was insufficient, and we sincerely apologize for any confusion caused by the expressions in the manuscript. To clarify, among the cases with measurable optical AL, one eye with keratoconus and cases undergoing concomitant vitrectomy (one eye with lens dislocation and one eye with zonule ligament rupture) were excluded from the analysis of predicted refractive error. We have revised the description in the Methods section accordingly to provide a clearer explanation of this point as follows. “Cases involving predicted refractive error in an eye with keratoconus (1 eye), concomitant vitrectomy (comprising 1 eye with lens dislocation, 1 eye with zonule ligament rupture) were excluded from the study.” (Methods, Page 8, Lines 85–86).

---

## [Decision Letter · Decision Letter 1]

26 Apr 2026

Impact of axial length correction in high intraocular pressure eyes on intraocular lens power calculation

PONE-D-25-53959R1

Dear Dr. Kojima,

We’re pleased to inform you that your manuscript has been judged scientifically suitable for publication and will be formally accepted for publication once it meets all outstanding technical requirements.

Kind regards,

Nader Hussien Lotfy Bayoumi, M.D., FRCS (Glasgow)

Academic Editor

PLOS One

Additional Editor Comments (optional):

Thank you for the elaborate responses to review comments

Reviewers' comments:

Reviewer's Responses to Questions

**Comments to the Author**

1. If the authors have adequately addressed your comments raised in a previous round of review and you feel that this manuscript is now acceptable for publication, you may indicate that here to bypass the “Comments to the Author” section, enter your conflict of interest statement in the “Confidential to Editor” section, and submit your "Accept" recommendation.

Reviewer #2: All comments have been addressed

2. Is the manuscript technically sound, and do the data support the conclusions?

Reviewer #2: Yes

3. Has the statistical analysis been performed appropriately and rigorously? 

Reviewer #2: I Don't Know

4. Have the authors made all data underlying the findings in their manuscript fully available?

Reviewer #2: Yes

5. Is the manuscript presented in an intelligible fashion and written in standard English?

Reviewer #2: Yes

6. Review Comments to the Author

Reviewer #2: No further comments.

By visual acuity notation, I meant which system was used i.e. decimal, Snellen, LogMAR, .. etc

7. PLOS authors have the option to publish the peer review history of their article (what does this mean?). If published, this will include your full peer review and any attached files.

Reviewer #2: No

---

## [Editor Report · Acceptance letter]

PONE-D-25-53959R1

PLOS One

Dear Dr. Kojima,

I'm pleased to inform you that your manuscript has been deemed suitable for publication in PLOS One. Congratulations! Your manuscript is now being handed over to our production team.

Kind regards,

on behalf of

Professor Nader Hussien Lotfy Bayoumi

Academic Editor

PLOS One